# Differential Role of Type 2 Diabetes as a Risk Factor for Tuberculosis in the Elderly versus Younger Adults

**DOI:** 10.3390/pathogens11121551

**Published:** 2022-12-16

**Authors:** Blanca I. Restrepo, Julia M. Scordo, Génesis P. Aguillón-Durán, Doris Ayala, Ana Paulina Quirino-Cerrillo, Raúl Loera-Salazar, America Cruz-González, Jose A. Caso, Mateo Joya-Ayala, Esperanza M. García-Oropesa, Alejandra B. Salinas, Leonardo Martinez, Larry S. Schlesinger, Jordi B. Torrelles, Joanne Turner

**Affiliations:** 1School of Public Health and UTHealth Consortium on Aging, University of Texas Health Science Center at Houston, Brownsville Campus, Brownsville, TX 78520, USA; 2School of Medicine, South Texas Diabetes and Obesity Institute, University of Texas Rio Grande Valley, Edinburg, TX 78541, USA; 3Host Pathogen Interactions and Population Health Program, Texas Biomedical Research Institute, San Antonio, TX 78227, USA; 4Barshop Institute, The University of Texas Health Science Center of San Antonio, San Antonio, TX 78229, USA; 5Secretaría de Salud de Tamaulipas, Reynosa 88630, Matamoros 87370 and Ciudad Victoria 87000, Mexico; 6Biology Department, University of Texas Rio Grande Valley, Edinburg, TX 78541, USA; 7Department of Health and Biomedical Sciences, University of Texas Rio Grande Valley, Edinburg, TX 78541, USA; 8Unidad Académica Multidisciplinaria Reynosa-Aztlán, Universidad Autónoma de Tamaulipas Reynosa-Aztlán, Reynosa 88779, Mexico; 9Department of Epidemiology, School of Public Health, Boston University, Boston, MA 02118, USA

**Keywords:** tuberculosis, elderly, diabetes, insulin resistance, inflammation, hyperglycemia, NSAID

## Abstract

The elderly are understudied despite their high risk of tuberculosis (TB). We sought to identify factors underlying the lack of an association between TB and type 2 diabetes (T2D) in the elderly, but not adults. We conducted a case–control study in elderly (≥65 years old; ELD) vs. younger adults (young/middle-aged adults (18–44/45–64 years old; YA|MAA) stratified by TB and T2D, using a research study population (n = 1160) and TB surveillance data (n = 8783). In the research study population the adjusted odds ratio (AOR) of TB in T2D was highest in young adults (AOR 6.48) but waned with age becoming non-significant in the elderly. Findings were validated using TB surveillance data. T2D in the elderly (vs. T2D in younger individuals) was characterized by better glucose control (e.g., lower hyperglycemia or HbA1c), lower insulin resistance, more sulphonylureas use, and features of less inflammation (e.g., lower obesity, neutrophils, platelets, anti-inflammatory use). We posit that differences underlying glucose dysregulation and inflammation in elderly vs. younger adults with T2D, contribute to their differential association with TB. Studies in the elderly provide valuable insights into TB-T2D pathogenesis, e.g., here we identified insulin resistance as a novel candidate mechanism by which T2D may increase active TB risk.

## 1. Introduction

The association between tuberculosis (TB) and type 2 diabetes mellitus (T2D) has been known for centuries but disappeared from the literature after the 1950s with the advent of insulin to treat T2D and discovery of effective first-line anti-TB drugs [1]. With the exponential growth of obesity over the past 30 years and associated increase in T2D, especially in countries where TB is endemic, the TB-T2D association is again at the frontline of risk factors for TB, with an estimated worldwide population attributable fraction of 34.5% [2,3]. T2D not only increases TB risk 3-fold, but also TB treatment failure including death [4].

Old age is a risk factor for T2D (~25%) amongst the elderly [5], as well as TB disease and subsequent death [6]. Thus, one would expect that T2D contributes to the higher TB morbidity and mortality in the elderly. While the association between TB and T2D is well established among all adults [2], this is not as apparent in the elderly given the lack of studies explicitly addressing this age difference [7,8,9]. At the Texas-Mexico border, we reported a 3-fold higher risk of TB among adult T2D patients [10], with an overall population attributable fraction (PAF) of TB due to T2D of 25%, being highest in middle-aged adults (48%). We recently expanded our studies to address the relationship between old age and TB, and while we found that nearly half of the elderly had T2D (47%), it was not associated with TB (OR 1.07, 95%CI 0.54, 2.12) [11].

A differential interaction between TB and T2D with age is not unique to TB. There is literature on age-associated attenuation in the risk of diseases due to changes in pathophysiology, metabolism, comorbidities and selective survival that modify risks for other diseases as well [12]. While multiple factors may be at play, a plausible contributor to the differential interaction between TB and T2D with age is the reported difference in T2D pathophysiology in elderly vs. young adults [13]. T2D in young adults is largely driven by obesity with low-grade inflammation and excess free fatty acids promoting insulin resistance leading to T2D [14,15]. Conversely, aging is associated with progressive loss in glucose regulation for every decade of life [13]. The underlying biology for T2D in old age is multifaceted and unclear, with influence from obesity, physical inactivity, body composition, diet and genetics [13]. Aging may also affect β-cell function with impaired response to incretins [13]. Given variations in underlying defects leading to T2D in the elderly, it is not unexpected that use of either fasting plasma glucose, hyperglycemia or HbA1c alone can miss a significant proportion of cases [13].

Understanding the differential relationship between TB, T2D and age will shed light on mechanisms underlying the higher risk of TB in young and middle-aged adults with T2D, but not the elderly, and provide new insights into TB pathogenesis more broadly. We posit that differences in the factors underlying T2D in young and middle-aged adults vs. elderly contribute to age-related differences in the association between TB and T2D. We evaluated the effect of age on the magnitude of the association between TB and T2D using our research study population and validating findings with TB surveillance data from Mexico. We then expanded our analysis in T2D patients only, to identify age-related differences in metabolic, sociodemographic and clinical features that associate with TB risk using multivariable logistic regression. Results indicate differences underlying glucose dysregulation and inflammation in elderly vs. young and middle-aged adults with T2D. We postulate mechanisms by which these factors can contribute to the waning TB-T2D association with older age.

## 2. Materials and Methods

### 2.1. Research Study Population and Age Categories

We enrolled adults ≥18 years old (y/o) with newly diagnosed pulmonary TB and non-TB controls. The latter consisted of close contacts (>5 h of shared airspace with new pulmonary TB case) or a convenience sample of community controls composed by a network of acquaintances of healthcare workers from the TB clinics with no recent history of exposure to a TB case, and complemented with participants attending a state-run adult daycare center to enrich for older individuals. Individuals with excessive alcohol intake, drug abuse or HIV were excluded since these are known risk factors for TB and are infrequent in elderly populations [11]. Participant characteristics were documented as described [11,16]. Briefly, T2D was based on hyperglycemia, chronic hyperglycemia (HbA1c ≥ 6.5%) or self-report [17]. Macrovascular and microvascular diseases, T2D medications and use of pain medications including non-steroidal anti-inflammatory drug (NSAIDs) was based on self-report. Laboratory testing included lipid profiles, complete blood counts, endogenous insulin (Mercodia), oxidized and reduced blood glutathione (Arbor Assays) and serum vitamin D (25-OH vitamin D direct, IBL International). Insulin resistance was estimated by HOMA1-IR in participants not using insulin [18]. Glycemic index was calculated by multiplying HbA1c by number of years with T2D.

### 2.2. Regional Registry for Tamaulipas, Mexico

TB surveillance data from new TB cases ± T2D in the state of Tamaulipas between 2006 and 2013 was used [19], but with exclusion of individuals with excessive alcohol consumption, intravenous drug use or HIV infection due to their age-independent higher risk of TB, and to match the exclusion criteria of our research study population [20,21]. The size and sex distribution of the reference general population was obtained from Tamaulipas census data for 2010 [22], and T2D prevalence by age and sex was obtained from the Mexican 2012 cross-sectional health assessment study, “Encuesta Nacional de Salud (ENSANUT)” [23].

### 2.3. Data Analysis

We conducted a case–control study to evaluate the association between TB and T2D in different age groups, and among T2D patients only to identify differences between the elderly vs. other adults. Data were analyzed using SAS vr. 9.4 (SAS Institute Inc., Cary, NC, USA). Age was the primary exposure with grouping of participants into young adults (YA, 18–44 y/o), middle-aged adults (MAA, 45–64 y/o) or elderly (ELD, ≥65 y/o). For some analysis MAA and YA were combined and referred to as ‘younger adults’. Chi-square or Fisher’s exact tests were used to compare categorical variables. Median values were compared by the Wilcoxon rank sum test. Comparisons of medians between more than two study groups were established by the Kruskall-Wallis test with post hoc Dwass, Steel, Critchlow-Fligner (DSCF) method. Multivariable logistic regressions were conducted to calculate adjusted odds ratios (AOR) of TB given the age group, after controlling for variables associated with TB across all age groups (*p* < 0.100 for sex, BMI, smoking) and exclusion of interrelated (collinear) variables (e.g., BMI with lipids); Interactions between age groups and T2D status were also considered. Given our low TB prevalence (<0.01%), crude and adjusted odds ratios (OR) can be considered rough estimations of relative risk in observational studies [24]. *p* values were considered significant if ≤0.05, and borderline significant if <0.10. Graphs were plotted using GraphPad PRISM vr. 9.0.(GraphPad Software, San Diego, CA, USA)

## 3. Results

### 3.1. Strength of the Association between TB and T2D by Age

T2D increases TB risk by 2- to 4-fold among adults [10,25], but we recently reported that T2D is not associated with TB among individuals ≥60 y/o vs. 18–50 y/o [11]. Here, we expanded this analysis in a case–control study that included a larger sample size, implementing an age cut-off of 65 y/o for the ELD group, and inclusion of adults of all ages. We studied 913 non-TB (455 YA, 333 MAA, 125 ELD) and 243 TB patients (105 YA, 97 MAA, 41 ELD) with participant characteristics summarized by age and TB status in Table 1 and Table 2. In the non-TB group (Table 1), the ELD had lower education, but more T2D, central obesity, abnormal cholesterols, macro- and micro-vascular diseases and NSAID use [11,26]. In the TB group (Table 2), the ELD had lower education or BCG vaccination, and higher central obesity or macrovascular and microvascular diseases [11,26]. Host factors associated with TB status in all age groups included sex, BMI, smoking and lipid levels (Appendix A). T2D was associated with TB in YA or MAA, but not the ELD by univariable analysis (Appendix A) and confirmed in multivariate models (Figure 1A; Appendix A). A significant interaction was found between the association TB-T2D in MAA vs. ELD (*p* = 0.024), but not YA vs. ELD (*p* = 0.191). Hyperglycemia or high HbA1c (≥6.5%) were also independently associated with higher odds of TB among YA or MAA, but not the ELD in multivariable models (Figure 1B,C). We further controlled for hemoglobin given its lower levels in the ELD (Appendix A) and its known influence on HbA1c [27], and found that high HbA1c remained positively associated with TB in YA/MAA, but not in the ELD (Figure 1D). Together, the association between TB and T2D or measures of glucose control was strongest in YA and waned with age to become non-significant in the ELD group.

### 3.2. Validation Using Data from the TB Control Program for Tamaulipas, Mexico

We validated our findings using data from the TB control program for the Mexican state of Tamaulipas 2006–2013 for TB statistics, after exclusion of individuals with HIV, drug abuse or excess alcohol to match our research study population (total n = 9661; adults n = 8783). For the general population, age and sex distribution was obtained from census data and T2D prevalence by age and sex from a Mexican cross-sectional health assessment report [23]. We confirmed published findings indicating that the incidence of TB increases with age, but the highest change was between children and YA/MAA, and with males having significantly higher risk in adults (Figure 2A) [28]. TB rates were higher in the ELD vs. children 0 to 19 y/o, or vs. all other age groups (≤64 y/o), but not vs. YA/MAAs (Figure 2B). In YA and MAAs, the prevalence of TB was higher in T2D vs. non T2D, particularly for males (Figure 2C), but this was not observed in the ELD group. In addition to relative risks estimates, absolute measures of risk also differed by age: Attributable fractions of TB among T2D patients waned with age: YA 82%, MAA 74% and ELD 27% for all participants, with similar results by sex (Figure 2D). The population attributable risk estimates were also lowest in the ELD vs. YA or MAAs. Sensitivity analysis with the original dataset prior to exclusion of individuals with HIV, drugs or alcohol excess revealed similar prevalence and attributable fractions (Appendix A). Together, findings from the Mexican TB control program are consistent with our study population, indicating a waning association between TB and T2D with age.

### 3.3. Differences in T2D Patients between ELD vs. YA/MAA

We next focused on T2D participants, to identify differences between ELD vs. YA/MAA that could explain their differential association with TB. First, we compared the co-occurrence of T2D-defining characteristics (e.g., hyperglycemia, high HbA1c, self-reported T2D) after controlling for potential confounders (sex, BMI, smoking, T2D medication use). Figure 3 illustrates the differences in the proportions of T2D-defining characteristics by univariable analysis, and significant differences in the ELD by multivariable analysis are indicated. Among T2D patients without TB, we found that co-occurrence of high glucose and HbA1c was lower in the ELD vs. YA (adj-*p* = 0.029) or MAA (adj-*p* = 0.072; Figure 3A). Instead, the ELD with T2D were more likely to have high HbA1c alone when compared to YA (adj-*p* = 0.010) and MAA (adj-*p* = 0.019).

Among T2D participants with TB, the ELD had a lower proportion of individuals with high blood glucose and HbA1c compared to YA (adj-*p* = 0.016) and MAA (adj-*p* = 0.007; Figure 3B). Instead, the ELD had higher HbA1c alone compared to YA (adj-*p* = 0.098) and MAA (adj-*p* = 0.076), or higher self-reported T2D alone when compared to YA (adj-*p* = 0.051) and MAA (adj-*p* = 0.039). There were no differences in the frequency of hyperglycemia alone between ELD and YA/MAA. Together, these findings suggested differences in the underlying pathophysiology of T2D in ELD vs. YA/MAA with TB.

To identify additional differences between ELD vs. YA/MAA in T2D patients, we studied those without TB since *M.tb* infection can modify glucose control [29]. ELD T2D patients had a higher proportion of females, lower education, and reported higher consumption of pain medications including NSAIDs (Table 3). They also had a longer history of T2D (ELD 10.0 years vs. YA 1.8 years or MAA 5.0 years), but their glycemia and HbA1c levels indicated better glucose control (Table 3 and Figure 4A). The glycemic index was higher in the ELD vs. YA/MAA (*p* = 0.025). Insulin levels did not differ by age, but the insulin resistance index was lower in the ELD (ELD 1.8 vs. YA 5.4 or MAA 3.5; *p* = 0.002). The ELD were more likely to use T2D medications (ELD 70.2% vs. MAA 68.2% or YA 51.7%; *p* = 0.056) and sulphonylureas in particular (ELD 48.9% vs. YA 18.3% or MAA 36.3%; *p* = 0.003) (Table 3; Appendix A). The ELD were less obese, but had more macrovascular diseases compared to YA/MAA (*p* ≤ 0.001) (Table 3). The ELD also had lower platelets, neutrophils and hemoglobin (Appendix A and Figure 4B). There were no differences between age groups in systemic glutathione oxidative status or vitamin D (Appendix A). Most of the unique features about the ELD remained significant after adjusting for sex and NSAID or sulphonylurea use (Appendix A). Together, these analyses revealed differences in socio-demographics, glucose control and T2D-associated characteristics between ELD vs. YA/MAA T2D participants.

### 3.4. Relationships between Level of Glucose Control and TB Risk

A notable difference between T2D in ELD vs. YA/MAA was lower severity in hyperglycemia and glycated hemoglobin. Thus, it is possible that T2D increases TB risk in any age group but only when hyperglycemia is severe. To test this hypothesis, we evaluated if the odds of TB would increase with higher cut-offs for hyperglycemia (125, 140 or 160 mg/dL) or HbA1c (6.5, 7.0 or 7.5%), regardless of the age group. After controlling for BMI and sex (and hemoglobin for HbA1c), results remained unchanged (Figure 5). Together, these findings did not support our hypothesis of a correlation between the magnitude of TB risk and level of glucose control.

## 4. Discussion

We showed that the magnitude of the TB/T2D association was highest in YA, with waning as age increased, and reaching a non-significant association for TB in the ELD group after controlling for confounders (Figure 1D). These findings were validated with notification data from TB patients reported to the Mexican state of Tamaulipas. This large dataset further revealed that absolute measures of TB risk in T2D patients decreased with age: from 82% in YA to 27% in ELD. Together, our findings provide strong support for the lower contribution of T2D to TB risk in the ELD group vs. YA/MAA. Additionally, our access to well-characterized participants in the research study population allowed for further exploration of differences between T2D patients in the ELD vs. YA and MAA groups, yielding insights into possible mechanisms by which T2D increases TB risk in YA/MAA only.

Understanding the underlying biology of the waning association between TB and T2D with aging can shed light into the mechanisms affecting their association. T2D in the elderly vs. adults has a different underlying pathophysiology that is not fully understood. Namely, in the elderly, T2D may be driven by slower insulin secretion in response to intestinal incretins, and insulin resistance may not be as prevalent, and if present it is mostly related to body composition (e.g., lower muscle mass for glucose uptake) and to physical inactivity [5]. In contrast, in younger adults, T2D is largely associated with obesity and peripheral insulin resistance [14]. Thus, we hypothesized that differences in T2D with aging may explain the reduction in TB risk with age. Accordingly, we found lower hyperglycemia or HbA1c levels in the ELD vs. YA/MAA. This milder glucose dysregulation in the ELD group may be due to their more prominent use of anti-glycemic agents (Table 3), or to delayed insulin secretion upon food intake that causes transient and mild post-prandial hyperglycemia [13]. Both scenarios are plausible. The latter mechanism is in line with our intriguing observation of a higher proportion of ELD diabetics having chronic hyperglycemia (high HbA1c), and yet, normal fasting glucose. We hypothesized that a less severe T2D in the ELD would explain their lack of association with TB, due to lower glucotoxicity [e.g., less oxidation and more advanced glycation end products (AGEs)] [30,31,32,33,34,35,36]. Thus, we expected increased rates of TB with higher cut-offs for hyperglycemia or HbA1c in the ELD. However, results did not support this hypothesis: There was no increase in TB rates with higher hyperglycemia or HbA1c in the ELD (nor MAA). These findings revealed two observations: First, in YA and MAA (but not ELD), the presence (vs. absence) of hyperglycemia defines the association between T2D and TB, but further deterioration in glucose control has no additional impact on the magnitude of TB risk. Second, in the elderly, hyperglycemia is not associated with TB risk. This led us to hypothesize that in the elderly there are events downstream of hyperglycemia that dampen glucotoxicity (oxidation, AGEs) and associated inflammation, that contribute to compromised immune containment of *M.tb* (Figure 6). To investigate possible mechanisms, we sought to identify additional differences between YA and MAA compared to ELD among participants with T2D.

We found that some reported features of the TB/T2D association in YA/MAA did not hold for the ELD. For example, in YA/MAA, TB usually develops after a multi-year history of T2D (~7 years [21]), while in the ELD we found a significantly longer T2D history (median 10 years) despite no association with TB. Sulfonylurea hypoglycemic drugs were used more frequently in the ELD. These drugs promote the development of anti-inflammatory M2-like macrophages in vitro and have been proposed as contributors to higher TB risk in T2D patients [37,38]. However, this assumption was not supported by our findings in the ELD with T2D. We evaluated other mechanisms by which T2D may increase TB risk, including higher oxidative stress that damages immune response molecules, or low levels of vitamin D that increase T2D risk and compromises responses to *M.tb* infection [30,39]. However, none differed between the ELD and YA/MAA with T2D.

Even though chronic low-grade inflammation is a hallmark of old age and T2D [40], the ELD vs. YA/MAA had features associated with less inflammation among people with T2D. For example, the ELD with T2D had lower prevalence of pro-inflammatory conditions like obesity and were more likely to take pain medications including NSAIDs or M2 macrophage-promoting sulfonylureas for glucose management [37]. The ELD also had lower neutrophils and platelets, which are associated with inflammation and bystander tissue damage in TB [41,42,43]. Reduced inflammation may also be beneficial in the ELD, e.g., NSAID use has been proposed as a host-directed therapy for TB [44], and in a randomized trial in TB patients with T2D, aspirin use significantly improved treatment outcomes [41]. Further, ibuprofen treatment of old mice resulted in decreased lung inflammation as measured in lung tissues, and also restored pulmonary macrophage responses when cells from ibuprofen treated old mice were isolated and infected with *M.tb* in vitro [45]. Thus, we posit that in the ELD with T2D there may be reduced inflammation that dampens the deleterious effect of hyperglycemia on compromised immune containment of *M.tb* (Figure 6). This could explain why hyperglycemia in the ELD is not associated with TB disease.

When compared to YA and MAA with T2D, the ELD with T2D had lower insulin resistance. It has been shown that monocytes are responsive to insulin with translocation of Glucose Transporter-4 (GLUT-4) to the cell membrane [46]; Other immune cells (e.g., lymphocytes, neutrophils) are also responsive to insulin, and insulin resistance compromises their function [47,48]. Therefore, it is plausible that higher insulin sensitivity in immune cells from ELD diabetics facilitates their use of glucose to meet the high energy demands required for an effective anti-*M.tb* response [49]. Higher insulin sensitivity in the ELD may also explain their lack of association between TB and T2D (Figure 6). We propose that in addition to the known deleterious effects of hyperglycemia on immune function, insulin resistance may be an additional mechanism by which T2D increases TB risk in younger adults.

We recognize study limitations. The diagnosis of T2D in the ELD may have been missed or underdiagnosed [13]. To increase sensitivity, we used a combination of diagnostic methods (HbA1c, fasting plasma glucose, or self-report) across all age groups [14]. In TB patients, we may have over-diagnosed T2D due to transient hyperglycemia resulting from inflammation [29,50], although the timing for testing was similar for all age groups. Contacts with subclinical incident TB were not identifiable in our study design; however, these cases are likely to be nondifferential by age. The size of the elderly groups may be underpowered to show significant differences by T2D status, and larger groups could show significant associations between TB and T2D in the elderly. However, this is not likely to affect our conclusions given that: (i) the strength of the association between T2D and TB decreases with older age, and (ii) T2D defining parameters like hyperglycemia and HbA1c are not likely to become significant even after increasing sample size. Our access to data from the entire Mexican state of Tamaulipas has the strength of thousands of records representing the entire population, but is limited by the timing of the dataset (2006–2013) and comparisons to the general population using indirect adjustments from census and T2D prevalence data [22,23]. Nevertheless, results were similar for the association between TB, T2D and old age with the surveillance dataset and our study population.

There are additional limitations inherent to studies in the elderly. The coexistence of multiple diseases besides T2D in old age could have contributed to negative confounding of the association between T2D and TB. There may have been survival bias of individuals more resistant to TB in the elderly; for example, there was a very low prevalence of smoking in the elderly, a factor known to increase excess mortality and TB risk [51,52]. We evaluated if our findings were due to ‘ceiling’ effects where the age-related decline in the strength of the association between TB and T2D could be due to an increasing prevalence of TB with old age (e.g., higher denominator resulting in lower relative risk estimates) [12]. However, we did not find strong support for this. First, in the Tamaulipas dataset the prevalence of TB did not show a significant increase with age [6], and hence, had negligible effect on the waning relative risk estimates of TB in old T2D participants. Second, absolute measures of TB risk in the elderly (Figure 2) resembled relative risk estimates, with both showing a waning relationship between T2D and TB with old age.

## 5. Conclusions

In summary, our findings indicate that the ELD vs. YA/MAA have different characteristics that define T2D (lower blood glucose and HbA1c), glucose uptake (lower insulin resistance), T2D management (use of more hypoglycemic agents), in an environment with less inflammation (less obese and use of anti-inflammatory drugs). We posit that a moderate and transient post-prandial hyperglycemia in the presence of insulin sensitivity, facilitates entry and use of glucose by monocytes and other immune cells, for more effective immune control of *M.tb*. In the ELD, this occurs in an environment where glucose levels are not high enough to induce oxidative damage of immune components [30]. Further studies are warranted to evaluate the impact of inflammation and insulin resistance as mechanisms driving TB risk in T2D patients. Given the high risk of ELD individuals of death from TB, it is not only ethical to study and understand risk factors for TB in this vulnerable group, but also valuable for the scientific community given the opportunity to reveal novel aspects of TB pathogenesis that are not evident in younger groups.

## Figures and Tables

**Figure 1 pathogens-11-01551-f001:**
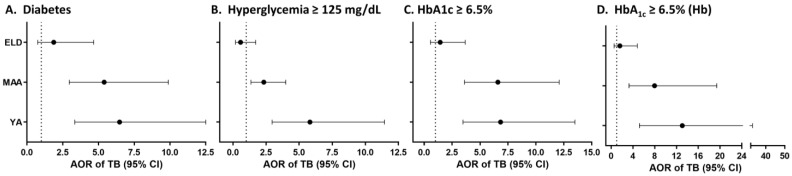
Association between TB and T2D, hyperglycemia or HbA1c by age groups among Texas-Mexico border participants. Analysis in YA, MAA or ELD or the adjusted OR of TB (vs. non-TB) given T2D, hyperglycemia or high HbA1c after controlling for core factors associated with TB in all age groups (sex, BMI, smoking), or these factors plus hemoglobin (Hb). Dots, odds ratio point estimates; horizontal lines, 95% confidence intervals. Dotted line for OR = 1.0 as reference; AOR, Adjusted OR; YA, 20–44y/o; MAA, 45–64 y/o; ELD, ≥65 y/o.

**Figure 2 pathogens-11-01551-f002:**
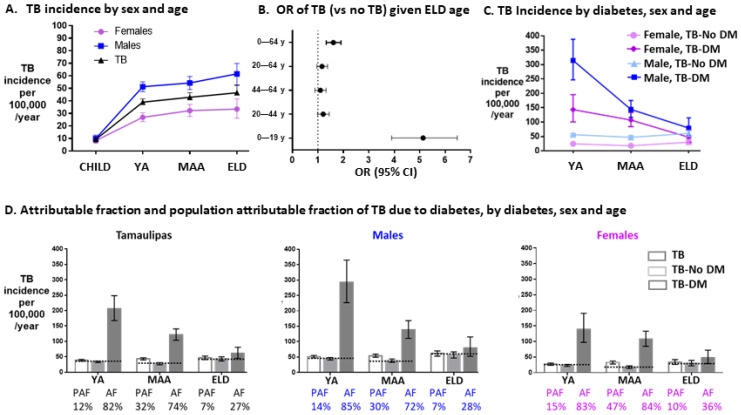
TB prevalence, risk and attributable fractions by age groups in Tamaulipas, Mexico 2006–2013. Analysis was conducted using the Mexican state of Tamaulipas TB surveillance datasets consisting of newly diagnosed TB patients between 2006 and 2013, and after exclusion of those consuming excessive alcohol, intravenous drugs or HIV-infected (n = 9661 total; 8783 adults). The total general population size, sex distribution and T2D prevalence by age was extrapolated from official state and national data. (**A**) Prevalence of TB by sex and age. (**B**) OR of TB vs. no TB in the ELD vs.: YA (OR 1.06, 95%CI 0.88, 1.27), MAA (OR 1.02, 95%CI 0.84, 1.24), all adults (OR 1.04, 95%CI 0.87, 1.24), or all age groups (OR 1.37, 95%CI 1.48, 1.63). (**C**) TB prevalence by T2D and sex, in YA, MAA and ELD age groups. (**D**) Relative risk, attributable fraction and population attributable fraction due to T2D, by sex and age group. Symbols and abbreviations: Dots, point estimates; Error bars, 95% confidence intervals; Horizontal dotted line, reference TB prevalence among individuals without T2D for calculation of T2D attributable fraction and population attributable fraction; RR, relative risk; PAF, T2D population attributable risk fraction for TB; AF, T2D attributable risk fraction for TB. CHILD, 0–19 y/o; YA, 20–44y/o; MAA, 45–64 y/o; ELD, ≥65 y/o.

**Figure 3 pathogens-11-01551-f003:**
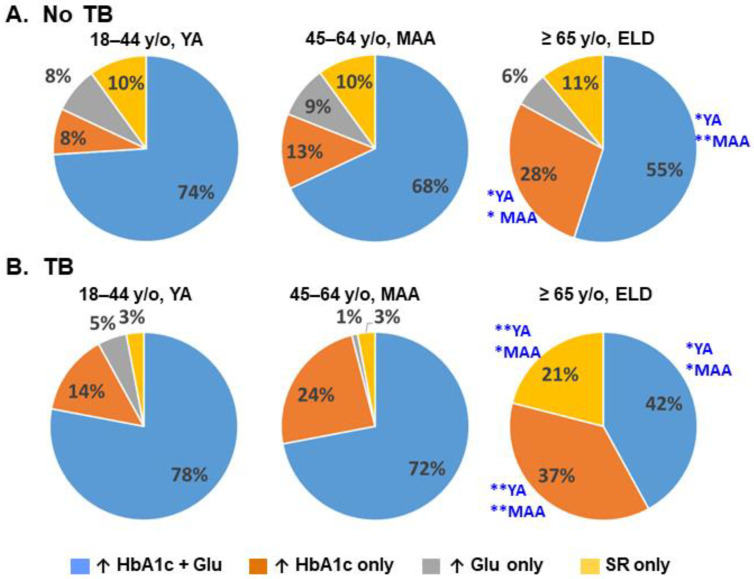
Differences in T2D-defining characteristics between ELD vs. YA/MAA. Among participants with T2D, there were differences in the distribution of T2D-defining characteristics (hyperglycemia, high HbA1c and/or self-reported T2D) in ELD vs. YA or MAA groups. Asterisks indicate significant (*****) or borderline significant differences (******) between ELD group and YA or MAA after controlling for age, sex, BMI, smoking and use of T2D medications; ↑, high values. SR, self-reported T2D; y/o, years old.

**Figure 4 pathogens-11-01551-f004:**
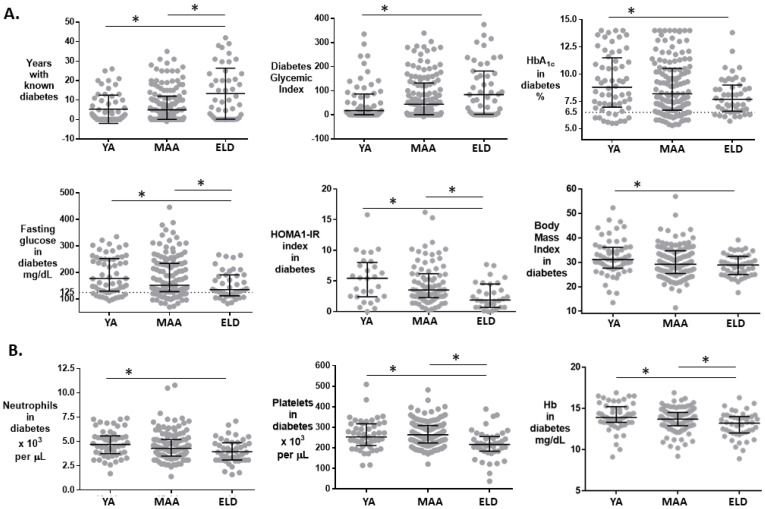
Characteristics of T2D participants by age groups. (**A**) Features of T2D history, glucose control, insulin resistance or BMI, by age groups. (**B**) Complete blood counts and differential (only significantly different features shown). Horizontal lines, median (IQR); *, *p* ≤ 0.099 vs. ELD group; dotted lines, cut-off for normal glycemia or HbA1c. Each dot represents a T2D participant without TB. Abbreviations: HOMA1-IR, homeostasis model assessment-estimated insulin resistance index; Hb, hemoglobin.

**Figure 5 pathogens-11-01551-f005:**
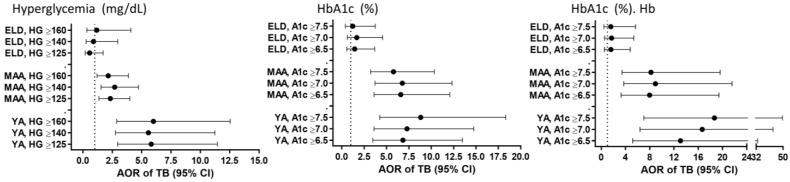
Adjusted OR of TB with different cut-offs for hyperglycemia or HbA1c, by age groups. Adjusted odds ratio of TB (vs. no TB) given fasting glucose hyperglycemia (≥125, 140 or 160 mg/dL) or high HbA1c (≥6.5, 7.0 or 7.5%) after controlling for BMI and sex (or hemoglobin for HbA1c) in YA, MAA and ELD participants. Dots, point estimates; horizontal lines, 95% confidence intervals. Dotted line indicates an OR = 1.0 as reference. AOR, Odds ratio adjusted for age, sex (and hemoglobin). HG, hyperglycemia; A1c, HbA1c; Hb, hemoglobin; YA, 20–44y/o; MAA, 45–64 y/o; ELD, ≥65 y/o.

**Figure 6 pathogens-11-01551-f006:**
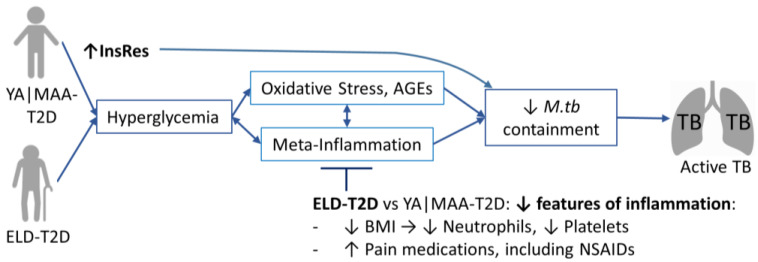
Hypothetical model for the waning association between TB and T2D in old age. The higher risk of TB in YA and MAA with T2D can be partly attributed to compromised containment of *M.tb* due to glucotoxicity (e.g., oxidative stress, AGEs) and inflammation. In this study we find that T2D is not associated with active TB in the ELD, and our results provide support for two non-mutually exclusive mechanisms that may explain this differential association when compared to younger adults. First, YA and MAA with T2D have higher insulin resistance when compared to the ELD, that can compromise immune function, and *M.tb* containment. Second, among T2D patients, the ELD have features of less inflammation compared to T2D in YA and MAA. The lower inflammation in the ELD may ameliorate the glucotoxic effects that hyperglycemia have on compromised *M.tb* containment. Abbreviations: YA|MAA-T2D, young or middle age adults with type 2 diabetes; ELD, elderly; InsRes, insulin resistance; AGEs, advanced glycation end products; *M.tb*, *M. tuberculosis*; BMI, body-mass index; NSAIDs, non-steroidal anti-inflammatory drugs.

**Table 1 pathogens-11-01551-t001:** Characteristics of participants by age groups, no TB.

	All (≥18 y/o)	YA (18–44 y/o)	MAA (45–64 y/o)	ELD (≥65 y/o)	YA vs. MAA vs. ELD
	n = 913	n = 455	n = 333	n = 125	*p* Value
**Sociodemographics**					
**Male sex**	294 (32.1%)	155 (33.8%)	106 (31.7%)	33 (26.4%)	0.263
**Education, High school or higher**	354 (38.6%)	**242 (52.8%)**	**99 (29.6%)**	13 (10.4%)	**<0.001**
**Current or past smoker**	260 (28.4%)	131 (28.6%)	96 (28.7%)	33 (26.4%)	0.868
**TB-related variables**					
**Recent TB exposure**	676 (73.7%)	**363 (79.3%)**	**246 (73.7%)**	67 (53.6%)	**<0.001**
**Past TB**	23 (2.5%)	11 (2.4%)	8 (2.4%)	4 (3.2%)	0.872
**Latent TB infection**	507 (55.3%)	**238 (52%)**	**191 (57.2%)**	78 (62.4%)	**0.093**
**BCG vaccination**	821 (89.5%)	420 (91.7%)	290 (86.8%)	111 (88.8%)	0.136
**Type 2 diabetes and other conditions**					
**Type 2 diabetes**	242 (26.4%)	**60 (13.1%)**	135 (40.4%)	47 (37.6%)	**<0.001**
**Overweight/obese, BMI ≥ 25**	705 (76.9%)	354 (77.3%)	263 (78.7%)	88 (70.4%)	0.184
**Central obesity (M ≥ 0.90 M; F ≥ 0.86)**	712 (77.6%)	**328 (71.6%)**	282 (84.4%)	102 (81.6%)	**<0.001**
**High cholesterol (200 mg/dL)**	153 (16.7%)	**42 (9.2%)**	85 (25.5%)	26 (20.8%)	**<0.001**
**High LDL (100 mg/dL)**	388 (42.3%)	**147 (32.1%)**	180 (53.9%)	61 (48.8%)	**<0.001**
**Low HDL (40 M, 50 F, mg/dL)**	655 (71.4%)	349 (76.2%)	221 (66.2%)	85 (68.0%)	**0.006**
**High Triglycerides (150 mg/dL)**	268 (29.2%)	113 (24.7%)	**117 (35.0%)**	38 (30.4%)	**0.006**
**Macrovascular diseases**	234 (25.5%)	**41 (9.0%)**	**114 (34.1%)**	79 (63.2%)	**<0.001**
**Microvascular diseases**	212 (23.1%)	**58 (12.7%)**	**98 (29.3%)**	56 (44.8%)	**<0.001**
**Anti-inflammatory medications**	171 (18.7%)	**66 (14.4%)**	**65 (19.5%)**	40 (32%)	**<0.001**

*p* values calculated by chi-square or Fisher’s exact. When comparison of the three age groups had *p* values ≤ 0.099, further comparisons were made for ELD vs. YA or MAA. Significant/borderline significant difference between YA or MAA vs. ELD are indicated by bold format of the percentages; *p* ≤ 0.099 shown in bold; Normal range values shown in parenthesis; M, males; F, females; N/A, not applicable.

**Table 2 pathogens-11-01551-t002:** Characteristics of participants by age groups, active TB.

	All (≥18 y/o)	YA (18–44 y/o)	MAA (45–64 y/o)	ELD (≥65 y/o)	YA vs. MAA vs. ELD
	n = 243	n = 105	n = 97	n = 41	*p* Value
**Sociodemographics**					
**Male sex**	151 (62.1%)	58 (55.2%)	67 (69.1%)	26 (63.4%)	**0.067**
**Education, High school or higher**	63 (25.9%)	**35 (33.3%)**	**25 (25.8%)**	3 (7.3%)	**0.006**
**Current or past smoker**	101 (41.6%)	37 (35.2%)	46 (47.4%)	18 (43.9%)	0.223
**TB-related variables**					
**Recent TB exposure**					N/A
**Past TB**	18 (7.4%)	8 (7.6%)	5 (5.2%)	5 (12.2%)	0.467
**Latent TB infection**					N/A
**BCG vaccination**	200 (82.3%)	**91 (86.7%)**	**81 (83.5%)**	28 (68.3%)	**0.025**
**Type 2 diabetes and other conditions**					
**Type 2 diabetes**	122 (50.2%)	**36 (34.3%)**	**67 (69.1%)**	19 (46.3%)	**<0.001**
**Overweight/obese, BMI ≥ 25**	63 (25.9%)	24 (22.9%)	**34 (35.1%)**	5 (12.2%)	**0.010**
**Central obesity (M ≥ 0.90 M; F ≥ 0.86)**	149 (61.3%)	**53 (50.5%)**	66 (68%)	30 (73.2%)	**0.013**
**High cholesterol (200 mg/dL)**	10 (4.1%)	4 (3.8%)	4 (4.1%)	2 (4.9%)	0.948
**High LDL (100 mg/dL)**	38 (15.6%)	17 (16.2%)	15 (15.5%)	6 (14.6%)	0.955
**Low HDL (40 M, 50 F, mg/dL)**	178 (73.3%)	81 (77.1%)	70 (72.2%)	27 (65.9%)	0.268
**High Triglycerides (150 mg/dL)**	22 (9.1%)	7 (6.7%)	13 (13.4%)	2 (4.9%)	0.221
**Macrovascular diseases**	53 (21.8%)	**13 (12.4%)**	**23 (23.7%)**	17 (41.5%)	**<0.001**
**Microvascular diseases**	88 (36.2%)	**24 (22.9%)**	49 (50.5%)	15 (36.6%)	**<0.001**
**Anti-inflammatory medications**	73 (30%)	31 (29.5%)	28 (28.9%)	14 (34.2%)	0.625

*p* values calculated by chi-square or Fisher’s exact. When comparison of the three age groups had *p* values ≤ 0.099, further comparisons were made for ELD vs. YA or MAA. Significant/borderline significant difference between YA or MAA vs. ELD are indicated by bold format of the percentages; *p* ≤ 0.099 shown in bold; Normal range values shown in parenthesis; M, males; F, females; N/A, not applicable.

**Table 3 pathogens-11-01551-t003:** Unique characteristics of type 2 diabetes patients without TB, by age groups.

	All (≥18 y/o)	YA (18–44 y/o)	MAA (45–64 y/o)	ELD (≥65 y/o)	YA vs. MAA vs. ELD
	n = 242	n = 60	n = 135	n = 47	*p* Value
**Sociodemographics and TB**					
**Male sex**	71 (29.5%)	**25 (41.7%)**	35 (26.1%)	11 (23.4%)	**0.054**
**Education, High school or higher**	70 (29.1%)	**29 (48.3%)**	**38 (28.4%)**	3 (6.4%)	**<0.001**
**Current smoker**	30 (12.4%)	9 (3.7%)	18 (7.4%)	3 (1.2%)	0.359
**NSAID use**	54 (24.2%)	**7 (13%)**	32 (26.0%)	15 (32.6%)	**0.057**
**BCG vaccination**	216 (89.3%)	53 (88.3%)	120 (89.6%)	43 (91.5%)	0.868
**Latent TB infection**	139 (57.7%)	32 (53.3%)	80 (59.7%)	27 (57.5%)	0.708
**Diabetes history**					
**Family history of diabetes**	197 (81.7%)	**52 (86.7%)**	111 (82.8%)	34 (72.3%)	0.145
**Self-reported diabetes**	179 (74.3%)	40 (66.7%)	103 (76.9%)	36 (76.6%)	0.298
**Years with diabetes**	4 (12.9)	**1.8 (9.3)**	**5 (11.9)**	10 (23.29)	**0.003**
**Glucose management**					
**Hyperglycemia, 2 levels at 125 mg/dL**	182 (75.5%)	**49 (81.7%)**	**104 (77.6%)**	29 (61.7%)	**0.041**
**HbA1c (%)**	8.1 (3.8)	**8.8 (4.3)**	8.2 (3.8)	7.7 (2.4)	0.106
**Glycemic Index (HbA1c * T2D Years)**	37.6 (125.6)	**17.8 (84.5)**	**43.9 (131.2)**	84.0 (179.8)	**0.025**
**Insulin Levels (mU/L)**	9.2 (12.0)	12.7 (13.5)	8.5 (10.9)	7.7 (12.8)	0.203
**HOMA-IR index**	3.3 (4.3)	**5.4 (5.4)**	**3.5 (3.8)**	1.8 (3.7)	**0.002**
**Diabetes medications in past month**					
**Any**	156 (65.2%)	**31 (51.7%)**	92 (68.2%)	33 (70.2%)	**0.056**
**Insulin**	38 (15.8%)	8 (13.3%)	20 (14.8%)	10 (21.3%)	0.487
**Metformin**	123 (51%)	24 (40.0%)	76 (56.3%)	23 (48.9%)	0.106
**Sulfonylureas**	83 (34.3%)	**18 (18.3%)**	49 (36.3%)	23 (48.9%)	**0.003**
**Metformin + Sulfonylureas**	65 (26.9%)	**9 (15%)**	40 (26.6%)	16 (34%)	**0.048**
**Diabetes-associated conditions**					
**Body-mass index**	29.6 (8.6)	**31.2 (8.5)**	29.2 (9.3)	28.9 (7.5)	**0.032**
**Central obesity (M ≥ 0.90 M; F ≥ 0.86)**	212 (88%)	**52 (86.7%)**	**123 (91.8%)**	37 (78.7%)	**0.012**
**Macrovascular diseases**	109 (45.2%)	**15 (25.0%)**	**61 (45.5%)**	33 (70.2%)	**<0.001**
**Microvascular diseases**	113 (46.9%)	26 (43.3%)	61 (45.5%)	26 (55.3%)	0.418

* Data expressed as column% for categorical variables or median (interquartile range, IQR) for continuous; Categorical variables compared by chi-square or Fisher’s exact tests, and medians by Kruskall-Wallis test with post hoc Dwass, Steel, Critchlow-Fligner (DSCF) method. When comparisons of the three age groups using chi-square or Fisher’s exact had *p* values ≤ 0.099, further comparisons were done for ELD vs. YA or MAA (Appendix A). Significant/borderline significant difference between YA or MAA vs. ELD are indicated by bold format of the percentages or medians; *p* values ≤ 0.099 shown in bold; NSAID, reported use of pain medications in past month, including non-steroidal anti-inflammatory drugs.

## Data Availability

Datasets are available upon request to the corresponding author.

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
