# Peer review of "Differential Role of Type 2 Diabetes as a Risk Factor for Tuberculosis in the Elderly versus Younger Adults"

_pathogens, 2022, doi:10.3390/pathogens11121551_

Round 1

Reviewer 1 Report

Thank you for sending me this manuscript. I found it of interest but have a number of comments and there are some important issues that require clarification

1. Study design. I find this some what unclear. The authors refer to a "research cohort" but I think the study design is a case-control study? This could be misleading and I suggest some terminology is revised. 

2. The choice of controls (non-TB controls) is extremely important in interpreting the results. The authors describe these are being from close contacts or community controls (convenience sample) and refer to 2 references for more details. I looked at reference 11 but don't see a description of how the controls were selected there. This should be expanded so the reader can judge the potential for selection bias. 

3. - There is a large literature in epidemiology on age attenuation of relative risks in epidemiology - e.g. Kaplan et al 1999. In itself this is therefore well known and not specific to TB-DM. See for example https://www.annualreviews.org/doi/10.1146/annurev.publhealth.20.1.89. There might be many reasons but often there are "ceiling effects" (e.g. very high rates of a bad outcome in the unexposed group at higher age making it difficult to identify a higher relative effect - in this case high DM prevalence in the controls) and possibly selection effects / survival bias / competing risks. As well as possibly differences in disease pathology at older ages (the argument they are making). I think this literature should be referenced and only the possibility of survival bias is mentioned briefly by the authors. In the discussion the authors need to be clearer that there are other plausible or competing explanations for their findings. 

4. Survivor bias also seems to me to be a possible explanation for the results. I noted the very low prevalence of smoking in the oldest age group - whilst this could be a birth cohort effect, it often suggests the presence of survivor bias (smokers having died younger) which could be discussed more fully. 

5. I would caution about how the results in the elderly groups are presented. Whilst I agree they seem to show age attenuation the sample size in the elderly group is actually quite small and it is therefore not surprising the effects shown were "non-significant" as they study is under-powered in this age group. 

6. There is a literature on age differences in the presentation of T2DM (e.g. differences by HbA1c and FPG) which it would be helpful to briefly summarise in the introduction. 

Author Response

R1 response to reviewers’ comments and suggestions:

We thank the reviewers for excellent suggestions that will provide the reader with a much better foundation for understanding unique features of individuals with old age, and how these may have implications for the interpretations of the findings from this study. Our detailed responses to each comment and suggestion are provided below.

REVIEWER 1

Comments and Suggestions for Authors

Thank you for sending me this manuscript. I found it of interest but have a number of comments and there are some important issues that require clarification:

  1. Study design. I find this somewhat unclear. The authors refer to a "research cohort" but I think the study design is a case-control study? This could be misleading, and I suggest some terminology is revised

Correct. The design is a case-control as indicated in the methods. The word cohort can also be used to refer to a group of individuals with shared characteristics.  However, since this may be misleading, we have replaced it with “study population” or “group” throughout the paper.

  1. The choice of controls (non-TB controls) is extremely important in interpreting the results. The authors describe these are being from close contacts or community controls (convenience sample) and refer to 2 references for more details. I looked at reference 11 but don't see a description of how the controls were selected there. This should be expanded so the reader can judge the potential for selection bias

Agree. An extended description of the enrollment criteria has been added and references have been removed.

Methods, Lines 89-95: “We enrolled adults ³18 years old (y/o) with newly diagnosed pulmonary TB and non-TB controls. The latter consisted of close contacts (>5h of shared airspace with new pulmonary TB case) or a convenience sample of community controls composed by a network of acquaintances of healthcare workers from the TB clinics with no recent history of exposure to a TB case, and complemented with participants attending a state-run adult daycare center to enrich for older individuals.”

  1. There is a large literature in epidemiology on age attenuation of relative risks in epidemiology - e.g. Kaplan et al 1999. In itself this is therefore well known and not specific to TB-DM. See for example https://www.annualreviews.org/doi/10.1146/annurev.publhealth.20.1.89. There might be many reasons but often there are "ceiling effects" (e.g. very high rates of a bad outcome in the unexposed group at higher age making it difficult to identify a higher relative effect - in this case high DM prevalence in the controls) and possibly selection effects / survival bias / competing risks. As well as possibly differences in disease pathology at older ages (the argument they are making). I think this literature should be referenced and only the possibility of survival bias is mentioned briefly by the authors. In the discussion the authors need to be clearer that there are other plausible or competing explanations for their findings. 

Thank you for bringing this up.

  1. Regarding differences in disease pathology between younger and older diabetics, we have done the following edits:
  • 3rd paragraph. Added sentence. Lines 70-73. Aging may also affect β-cell function with impaired response to incretins [13]. Given variations in underlying defects leading to T2D in the elderly, it is not unexpected that use of either fasting plasma glucose, hyperglycemia or HbA1c alone can miss a significant proportion of the cases [13].”
  • 2nd paragraph. Lines 296-304. Added text: “Understanding the underlying biology of the waning association between TB and T2DM with aging can shed light into the mechanisms affecting their association. T2DM in the elderly vs adults has a different underlying pathophysiology that is not fully understood. Namely, in the elderly, T2DM may be driven by slower insulin secretion in response to intestinal incretins, and insulin resistance may not be as prevalent, and if present it is mostly related to body composition (e.g. lower muscle mass for glucose uptake) and to physical inactivity [5]. In contrast, in younger adults, T2DM is largely associated with obesity and peripheral insulin resistance [14]. Thus, we hypothesized that differences in T2DM with aging may explain the reduction in TB risk with age. Accordingly, we found…”
  1. Regarding study limitations and caution with interpretation of findings in studies done with elderly adults, we have done the following edit:
  • Lines 396-408: New paragraph discusses possible study limitations inherent to elderly studies, and how these are applicable to our study. We added new concepts, including the possibility of ceiling effects. “There are additional limitations inherent to studies in the elderly. The coexistence of multiple diseases besides T2D in old age could have contributed to negative confounding of the association between T2D and TB. There may have been survival bias of individuals more resistant to TB in the elderly; for example, there was a very low prevalence of smoking in the elderly, a factor known to increase excess mortality and TB risk [51,52]. We evaluated if our findings were due to ‘ceiling’ effects where the age-related decline in the strength of the association between TB and T2D could be due to an increasing prevalence of TB with old age (e.g. higher denominator resulting in lower relative risk estimates) [12]. However, we did not find strong support for this. First, in the Tamaulipas dataset the prevalence of TB did not show a significant increase with age [6], and hence, had negligible effect on the waning relative risk estimates of TB in old T2D participants. Second, absolute measures of TB risk in the elderly (Fig. 2) resembled relative risk estimates, with both showing a waning relationship between T2D and TB with old age.”
  1. Survivor bias also seems to me to be a possible explanation for the results. I noted the very low prevalence of smoking in the oldest age group - whilst this could be a birth cohort effect, it often suggests the presence of survivor bias (smokers having died younger) which could be discussed more fully. 

Yes. We bring this up in the new paragraph discussing possible limitations in the Discussion (see response 3). 

I would caution about how the results in the elderly groups are presented. Whilst I agree they seem to show age attenuation the sample size in the elderly group is actually quite small and it is therefore not surprising the effects shown were "non-significant" as they study is under-powered in this age group. 

Yes. A potential limitation of the elderly groups is their relatively small sample size, and enrollment of more participants may result in a significantly higher relative risk of TB in the elderly group with T2D. However, this finding would not affect the biological implications of our current findings because: i) The point estimates for diabetes are lower with increasing age; ii) with the current sample size we can detect an interaction effect between age and the association between T2D and TB; and ii) the point estimates for diabetes defining parameters like hyperglycemia and HbA1c do not indicate increase odds ratios of TB in T2D; In fact, the point estimate for hyperglycemia suggests possible protection. We have added these comments to the study limitations in the Discussion section.

Discussion. Lines 384-389. New text: “The size of the elderly groups may be underpowered to show significant differences by T2D status, and larger groups could show significant associations between TB and T2D in the elderly. However, this is not likely to affect our conclusions given that: i) the strength of the association between T2D and TB decreases with older age, and ii) T2D defining parameters like hyperglycemia and HbA1c are not likely to become significant even after increasing sample size.”  

  1. There is a literature on age differences in the presentation of T2DM (e.g. differences by HbA1c and FPG) which it would be helpful to briefly summarize in the introduction. 

Yes. We have added a sentence to the Introduction to expand on this. It is actually relevant because it provides further foundation for our findings in Figure 3 and our discussion on the possible limitations in the diagnosis of T2D in the elderly.  

Introduction. Lines 71-73.  “Given variations in underlying defects leading to T2D in the elderly, it is not unexpected that use of either fasting plasma glucose, hyperglycemia or HbA1c alone can miss a significant proportion of cases [13].”

Reviewer 2 Report

The article is about the influence of T2D on TB disease. Recently, the WHO made an alert about tuberculosis and its damage to the global population. Considering this health alert and the problem with immunization against M. tuberculosis worldwide, this article is fascinating and can contribute to the anamnesis of patients with T2D in developing countries. 

However, a few details to consider:

  1. The figure abstract and the hypothetical model try to have the same nomenclature and information for the age groups.
  2. The order of the figures has to be according to the text (Fig 4 in section 5).
  3. In the discussion, page 11; lines 337-338, the authors mentioned a study in vivo about inflammation and ibuprofen. However, then the statement finished with results in vitro. Please fix and explain this a little more of this study.
  4. Update the reference from the 90´s.
  5. The number of pages are not ordered correctly.

Author Response

REVIEWER # 2

The article is about the influence of T2D on TB disease. Recently, the WHO made an alert about tuberculosis and its damage to the global population. Considering this health alert and the problem with immunization against M. tuberculosis worldwide, this article is fascinating and can contribute to the anamnesis of patients with T2D in developing countries. 

However, a few details to consider:

  1. The figure abstract and the hypothetical model try to have the same nomenclature and information for the age groups.

My apologies. I do not understand this suggestion.

  1. The order of the figures has to be according to the text (Fig 4 in section 5).

Done. I moved figure 4 to section 4 which is where it is cited, although there is a lot of blank space left, but I am open as to how the space is accommodated.

  1. In the discussion, page 11; lines 337-338, the authors mentioned a study in vivo about inflammation and ibuprofen. However, then the statement finished with results in vitro. Please fix and explain this a little more of this study.

Yes. We have edited to clarify. It now reads:

Discussion lines 347-349. “Further, ibuprofen treatment of old mice resulted in decreased lung inflammation as measured in lung tissues, and also restored pulmonary macrophage responses when cells from ibuprofen treated old mice were isolated and infected with M.tb in vitro [45].”

  1. Update the reference from the 90´s.

We have updated two references from the 1990s to one from 2018, but left two that remain relevant in contemporary times: Kaplan et al, 1999 provide an excellent overview on the changes in magnitude of risk factors with age; Kim et al., 1995 conducted one of the few studies where the association between TB and T2D is broken down by age groups.

  1. The number of pages are not ordered correctly.

It was quite a challenge to work with the formatting.  I would request the editorial office to provide support with this aspect. Thank you in advance.